# Recent Advances in Urinary Peptide and Proteomic Biomarkers in Chronic Kidney Disease: A Systematic Review

**DOI:** 10.3390/ijms24119156

**Published:** 2023-05-23

**Authors:** Lorenzo Catanese, Justyna Siwy, Harald Mischak, Ralph Wendt, Joachim Beige, Harald Rupprecht

**Affiliations:** 1Department of Nephrology, Angiology and Rheumatology, Klinikum Bayreuth GmbH, 95447 Bayreuth, Germany; lorenzoriccardo.catanese@gmail.com (L.C.); harald.rupprecht@klinikum-bayreuth.de (H.R.); 2Kuratorium for Dialysis and Transplantation (KfH), 95445 Bayreuth, Germany; 3Medizincampus Oberfranken, Friedrich-Alexander-University Erlangen-Nürnberg, 91054 Erlangen, Germany; 4Mosaiques Diagnostics GmbH, 30659 Hannover, Germany; siwy@mosaiques-diagnostics.com; 5Department of Nephrology, St. Georg Hospital Leipzig, 04129 Leipzig, Germany; ralph.wendt@sanktgeorg.de (R.W.); joachim.beige@kfh.de (J.B.); 6Department of Internal Medicine II, Martin-Luther-University Halle/Wittenberg, 06108 Halle/Saale, Germany; 7Kuratorium for Dialysis and Transplantation (KfH), 04129 Leipzig, Germany

**Keywords:** biomarkers, chronic kidney disease, peptide, proteomic, urine

## Abstract

Biomarker development, improvement, and clinical implementation in the context of kidney disease have been a central focus of biomedical research for decades. To this point, only serum creatinine and urinary albumin excretion are well-accepted biomarkers in kidney disease. With their known blind spot in the early stages of kidney impairment and their diagnostic limitations, there is a need for better and more specific biomarkers. With the rise in large-scale analyses of the thousands of peptides in serum or urine samples using mass spectrometry techniques, hopes for biomarker development are high. Advances in proteomic research have led to the discovery of an increasing amount of potential proteomic biomarkers and the identification of candidate biomarkers for clinical implementation in the context of kidney disease management. In this review that strictly follows the PRISMA guidelines, we focus on urinary peptide and especially peptidomic biomarkers emerging from recent research and underline the role of those with the highest potential for clinical implementation. The Web of Science database (all databases) was searched on 17 October 2022, using the search terms “marker *” OR biomarker * AND “renal disease” OR “kidney disease” AND “proteome *” OR “peptid *” AND “urin *”. English, full-text, original articles on humans published within the last 5 years were included, which had been cited at least five times per year. Studies based on animal models, renal transplant studies, metabolite studies, studies on miRNA, and studies on exosomal vesicles were excluded, focusing on urinary peptide biomarkers. The described search led to the identification of 3668 articles and the application of inclusion and exclusion criteria, as well as abstract and consecutive full-text analyses of three independent authors to reach a final number of 62 studies for this manuscript. The 62 manuscripts encompassed eight established single peptide biomarkers and several proteomic classifiers, including CKD273 and IgAN237. This review provides a summary of the recent evidence on single peptide urinary biomarkers in CKD, while emphasizing the increasing role of proteomic biomarker research with new research on established and new proteomic biomarkers. Lessons learned from the last 5 years in this review might encourage future studies, hopefully resulting in the routine clinical applicability of new biomarkers.

## 1. Introduction

Chronic kidney disease (CKD) is one of the most challenging global health burdens in present time and has a severe impact on the morbidity and mortality of western societies [1]. The guidelines from the global organization Kidney Disease Improving Global Outcomes (KDIGO) regarding the evaluation and management of CKD are currently being updated after the comprehensive guidelines published in 2012 [2]. The diagnosis and management of CKD have been linked to a handful of well-established and routinely assessed biomarkers, including serum creatinine and more specifically the creatinine-derived and -calculated estimated glomerular filtration rate (eGFR), as well as urinary albumin excretion or the urine albumin creatinine ratio (UAE and UACR). After decades of scientific evidence and clinical experience using these biomarkers, they have become a valuable tool for physicians and scientists. However, these biomarkers have well known limitations and shortcomings. Creatinine levels have interindividual variances, depend on other factors such as muscle mass, and often only rise when significant kidney function has already been lost [3]. To minimize the effect of these variabilities in order to optimize the estimation of the eGFR, over 70 equations accounting for sex, ethnicities, and disease entities have been proposed over the last decade, and yet eGFR has not been able to reach the accuracy of measured GFR, which never has become a routine biomarker due to its limited applicability [4]. Albuminuria finds a broad usage in the monitoring and guiding of therapeutical decisions in the context of diabetes and diabetic nephropathy [5], but it is far from an ideal biomarker for CKD due to its high variability, even in measurements within the same individual, and its low specificity when a diagnosis of kidney disease has not yet been established [6,7,8]. Therefore, finding the correct diagnosis of CKD, predicting the disease’s progression, and the guidance of therapeutic decisions still may require the performance of a kidney biopsy and histopathological analysis. Using histopathological biomarkers, such as the extent of renal interstitial fibrosis, can significantly improve the prediction of the disease progression and many therapeutical decisions, especially in the context of glomerular disease, which strongly relies on kidney biopsies. However, in day to day clinical business, biopsies are often not performed or are unavailable due to the invasive nature of obtaining specimens, the associated risks, and contraindications [9]. Hence, the need for better biomarkers has risen and has been the subject of scientific research in the CKD area over the last decades. A comprehensive review regarding single peptide non-invasive biomarkers has been published in the past [10]. Taking into account the complex pathogenesis of kidney disorders, multi-peptide approaches have become a promising approach in biomarker discovery and proteomic research has now been fairly well established in the nephrological community [11]. Serum and urine provide optimal sources for mass-spectrometry-coupled proteomics due to their broad availability in clinical routine. The collection of urine is entirely non-invasive and urinary proteomics might allow for more precise insights, due to the obvious direct link to kidney conditions. The detection of molecular changes at the proteome level may enable the timely detection of the disease prior to irreversible organ damage, allow for early and appropriate therapy that is available today, and may such prolong patients’ life and quality of life. (Figure 1). In this review, we aim to give a comprehensive overview of the protein and peptide biomarkers of CKD that have been discovered within the last 5 years, while emphasizing the rising importance of proteomic biomarkers in the diagnosis, prediction of progress, and therapy of CKD.

## 2. Methods/Search Criteria

A review of the recent literature regarding the urinary biomarkers of CKD was performed, strictly following the PRISMA guidelines and using the Web of Science database (all databases). Manuscripts published from October 2017 until October 2022 were considered eligible for this review. Titles and topics were screened for the search terms “marker *” OR biomarker * AND “renal disease” OR “kidney disease” AND “proteome *” OR “peptid *” AND “urin *”. The search strategy is presented in Figure 2. The initial search resulted in 3669 hits with 1197 manuscripts cited more than 5 times per year. Review articles and conference abstracts were excluded. Publications from the year 2022 with less than 5 citations per year were manually selected according to their relevance to this review. Furthermore, articles based on animal models, metabolites, and kidney transplants were excluded. Additionally, studies focusing on micro-RNA and exosomal vesicles were excluded. This resulted in a total of 633 papers, which were examined by the authors. These publications and publications from the year 2022 with less than 5 citations per year were manually selected according to their relevance to this review. Finally, a total of 62 manuscripts, listed in the Appendix A, served as a basis for this review.

## 3. Results

### 3.1. Uprising Single-Protein Urine Biomarkers of Chronic Kidney Disease

The numerously cited research articles from the last 5 years revealed a collection of single protein-based biomarkers for kidney diseases and disease progression. Most of them were established in prior years and are still the focus of current investigations, highlighting their importance for the monitoring of CKD.

#### 3.1.1. CD80

Nephrotic syndrome is the second most common cause of CKD in the first three decades of life. Its underlying pathologies move on a spectrum of diseases and a correct diagnosis is pivotal for an estimation of its prognosis and therapy, as some are steroid-sensitive and others are not. There has been ongoing discussion about the differentiation of minimal change disease (MCD) and focal segmental glomerular sclerosis (FSGS), which may be manifestations of the same pathomechanism at different stages [12]. Discrimination through kidney biopsy findings is, to some extent, possible, but often remains inconclusive [12]. Thus, hope for a biomarker-based differentiation has risen. Gonzalez Guerrico et al. [13] studied a large cohort of 411 patients with different causes of nephrotic syndrome. An ELISA-based measurement of CD80 was performed on urine samples from the patients. They found the CD80 levels to be significantly higher in the MCD patients than those in any other groups and also a significant increase in CD80 in active FSGS and MCD. They concluded CD80 to be a discriminator of MCD from other forms of nephrotic syndrome, especially secondary FSGS. In a pediatric study, 64 patients with nephrotic syndrome were evaluated for their urinary CD80 levels. Here, patients with high urinary CD80 had a good response to immunosuppressive therapy and a significantly lower risk of progressing to CKD, possibly underlining the differentiation between MCD and FSGS [14]. CD80 has previously been suggested for the differentiation of MCD and FSGS [15,16]. 

#### 3.1.2. Dickkopf-Related Protein 3

Dickkopf-related protein 3 (DKK3) is a secreted glycoprotein derived from tubular epithelia cells. Its involvement in the canonical WNT-β-Catenin signalling pathway has been shown to be a potential driver of kidney fibrosis, a hallmark of progressing CKD [17,18]. DKK3 may have potential as a urinary biomarker for kidney disease because it is secreted into the urine under tubular stress. In a prospective cohort of 351 patients with CKD stages two and three, Sánchez-Álamo et al. showed that the urinary DKK3 to creatinine ratio was significantly higher in patients that reached the primary composite outcome: a 50% increase in serum creatinine, end-stage kidney disease (ESKD), or death [19]. The uDKK3 levels correlated with the baseline proteinuria and subsequently rose in subjects with higher proteinuria. Treatment with RAS-blockers did not affect the uDKK3 levels. In another prospective cohort of 575 patients with CKD stages two–four, with various underlying CKD etiologies and 481 healthy controls, the baseline urinary DKK3 to creatinine ratio was shown to be significantly higher in the CKD group than that in the healthy population. In the CKD cohort, a urinary DKK3 to creatinine ratio of >4000 pg/mg was associated with an annual eGFR decline of 7.6%, and its predictive properties were superior to eGFR and albuminuria alone. Furthermore, uDKK3 levels were correlated with the degree of tubulointerstitial fibrosis [20]. Another study examined the preoperative uDKK3 levels in patients undergoing cardiac surgery. In 471 patients, the DKK3 to creatinine ratios of >471 pg/mg were predictive of short-term acute kidney injury (AKI), persistent kidney impairment, and dialysis dependency [21]. These three studies showed evidence for DKK3 as a urinary biomarker independent of the underlying CKD etiology. Two further studies included in this review focused on DKK3 as a biomarker in the context of contrast-mediated (CM) kidney disease. Historically, CM nephropathy has been a widely accepted phenomenon with ongoing discussions and controversies [22]. In one study comprising 490 patients undergoing coronary angiography, their uDKK3 levels were assessed 24 h prior to the procedure. The uDKK3 baseline levels were significantly higher in individuals that developed AKI. However, the follow-up uDKK3 levels were not higher in individuals that developed AKI [23]. In a second study on 458 patients scheduled for cardiovascular procedures requiring contrast medium administration, the baseline uDKK3 to creatinine ratio was predictive for the development of AKI and persistent kidney dysfunction, with thresholds of >491 pg/mg and >322 pg/mg, respectively [24]. As reflected by the multiple studies presenting very different thresholds for disease detection, more detailed investigations are necessary to allow for meaningful clinical implementation.

#### 3.1.3. Epidermal Growth Factor

Epidermal growth factor (EGF) is a tubule-specific kidney polypeptide which confers biological functions such as cellular metabolism and glomerular hemodynamics, cell growth, and injury repair [25]. While EGF is absent in plasma samples, urinary EGF excretion is a physiological phenomenon in healthy individuals. Several recent studies have found decreasing levels of urinary EGF to be associated with several kidney diseases and progressive kidney damage. In a study on 1032 patients with type 2 diabetes and normal kidney function, several single peptide biomarkers were assessed for their predictive value for early kidney function decline in a 5–12-year follow-up period. Urinary EGF and the EGF to MCP-1 ratio were significantly associated with the risk of early kidney function decline and a combination of all these markers resulted in a significant improvement in the predictive performance regarding early kidney function decline [26] In a cross-sectional study, 1811 patients with early-stage diabetic kidney disease (DKD) and type 2 diabetes patients without DKD and 208 patients with advanced-stage DKD were included. The urinary EGF to creatinine ratio (uEGF/Cr) was positively correlated with eGFR and negatively correlated with the occurrence of DKD (OR 0.65; *p* < 0.001). In the longitudinal observation of the advanced DKD cases, the uEGF/Cr was associated with a percentage change in the eGFR slope, a composite endpoint of ESKD, and a 30% reduction in eGFR [27]. Menez et al. measured the uEGF levels in 865 patients after cardiac surgery. In addition to the urinary biomarker study, a tissue transcriptomic analysis was performed. The authors studied patients with and without clinically apparent CKD and found that higher levels of uEGF were protective with regard to a complex composite outcome of the incidence and progression of CKD [28]. In a Norwegian and Dutch cooperation, patients from the RENIS and PREVEND cohorts were recruited and investigated for their urinary EGF levels. The study populations included individuals without diabetes or CKD and kidney function was assessed using iohexol-measured GFR in the RENIS cohort and CKD-EPI-based eGFR in the PREVEND cohort. After adjustments for GFR, the ACR in urine, and CKD risk factors, lower uEGF levels were associated with a rapid GFR loss in both cohorts and a lower uEGF was associated with incident CKD in a combined analysis [29]. In a smaller study on 83 patients with DKD, the authors investigated, among others uEGF/Cr. The primary outcome was defined as an eGFR loss of more than 25% per year. During a follow-up time of 23 months, patients with a rapid eGFR decline showed significantly lower levels of uEGFR/Cr. Other biomarkers were also tested for their predictive value in eGFR decline and none were superior to the classic marker UACR [30]. In a pediatric study on 117 patients with Alport syndrome and 146 healthy children, uEGF/Cr was inversely correlated with eGFR. Moreover, it was found that uEGF/Cr was inversely associated with aging and a more rapid eGFR decline was observed in children with Alport syndrome. A longitudinal follow-up was available for 38 children. In these patients, there was a significant correlation between uEGF/Cr and the eGFR slope (r = 0.58, *p* < 0.001), and the predictive value of uEGF/Cr was superior to eGFR or proteinuria, with an AUC of 0.88 vs 0.77 and 0.81, respectively. These findings show promise for uEGF as a progression marker for CKD and especially DKD [31].

#### 3.1.4. Kidney Injury Molecule 1

Kidney injury molecule 1 (KIM-1) is a membrane protein expressed in the liver, spleen, and kidney. It has been shown to play a role in kidney disease and kidney injury through a number of different molecular targets and serve as a biomarker for AKI and CKD [32]. In a study on 602 patients with type 2 diabetes, their serum and urinary KIM-1 levels were assessed and found to be correlated with UACR. However, only serum KIM-1 was associated with eGFR [33]. In the early stages of DKD, urinary KIM-1 showed an association with higher incidences of albuminuria and also progressions of albuminuria in a longitudinal observation [34]. Brunner et al. included an evaluation of 10 different urinary biomarkers in the context of lupus nephritis (LN). The biomarkers most closely and consistently associated with the histological scores of LN were adiponectin and osteopontin, though KIM-1 showed an association with eGFR decline and the histology of LN [35]. Another study on 257 patients with type 2 diabetes evaluating five different urine biomarkers showed a higher risk for rapid eGFR loss and progression to ESKD for the highest quartile of urinary KIM-1 among the population (hazard ratio (HR) 2.77, 95% CI, 1.27–6.05) [36]. In the previous mentioned study by Nowak et al., urinary KIM-1 was also found to be associated with an early decline in kidney function in type 2 diabetes patients [26].

#### 3.1.5. Monocyte Chemoattractant Protein-1 

Monocyte chemoattractant protein-1 (MCP-1) or CC-chemokine ligand 2 (CCL2) is a chemotactic cytokine that confers innate immunity and tissue inflammation through its role in monocyte/macrophage recruitment and migration. Several kidney cells, including mesangial cells and podocytes, have been shown to release MCP-1 after a variety of inflammatory stimuli and induce numerous inflammatory cascades [37]. In a large prospective multicenter study on 1538 hospitalized patients, several urinary proteins were assessed in the context of CKD progression. MCP-1 levels were correlated with a rapid loss of kidney function and associated with a higher incidence of the composite outcome encompassing CKD incidence, CKD progression, ESKD, and death [38]. The eGFR loss in the highest MCP-1 quartile was 17.8% (95% CI, 16.7–18.8), annually compared to 8.0% (95% CI, 7.1–9.0) in the lowest quartile. The HR for the association of the composite kidney outcome with MCP-1 levels was 1.32. In the earlier mentioned study by Wu et al., with evidence for uEGF/Cr as a potential biomarker for DKD, uMCP-1/Cr was also assessed, but no significant difference in uMCP-1/Cr was observed between diabetic patients with or without DKD. However, a significant correlation between uMCP-1/Cr and the extent of albuminuria was found and both the uMCP-1/Cr and uEGF/MCP-1 ratios were independently associated with the composite kidney endpoint [27]. In another study on DKD with 83 patients, rapid progressors had higher levels of uMCP-1 and lower uEGF and uEGF/uMCP-1 ratios. A prediction of the composite outcome showed an area under the receiver operating characteristic curve (ROC-AUC) of 0.73 and 0.74 for uMCP-1 and uEGF/uMCP-1, respectively. In contrast to uEGF alone, uMCP-1 and uEGF/uMCP-1 were independently associated with rapid eGFR decline in a multivariate analysis [30]. In the above mentioned study on cardiac surgery patients, uMCP-1 levels were also independently positively associated with the composite CKD outcome, with an HR of 1.10 [28]. Three studies focused on MCP-1 in patients with systemic lupus erythematodes (SLE) and LN, highlighting the potential role of MCP-1 as a disease-specific biomarker for kidney involvement in SLE. In a study on 197 Caucasian SLE patients, a panel of six urinary biomarkers, including MCP-1, was assessed and compared to healthy controls (*n* = 48). The prediction of kidney involvement, as well as the treatment response to Rituximab, were tested. The uMCP-1 levels were higher in the SLE patients compared to those in the healthy controls, and MCP-1, among four other markers, was higher in patients with active LN compared to non-active LN. An ROC analysis using a combined biomarker, including MCP-1, showed an AUC of 0.898 for predicting LN. A different biomarker combination encompassing MCP-1 was predictive of the treatment response to Rituximab [39]. In a second study on 120 SLE patients, several urinary biomarkers were investigated for their correlations with the histological signs of kidney disease activity and chronic kidney damage in the biopsy specimen. A histopathological analysis was performed on 55 patients. uMCP-1 was higher in patients with chronic kidney involvement, but also patients with a crescent formation and higher levels of kidney fibrosis [35]. In a third study on 89 patients with childhood-onset SLE, uMCP-1 was analyzed among nine other urinary biomarkers and its correlation with the histological features of LN, as well as its correlation with a rapid loss of eGFR after 12 months, was investigated. In this study, the uMCP-1 levels did not correlate with the histological features of LN and there was no difference in the MCP-1 levels between kidney disease progressors and non-progressors [35]. In the above-mentioned study by Nowak et al., MCP-1, especially in combination with EGF, was shown to be associated with an early decline in eGFR in type 2 diabetes patients with initially normal kidney function [26]. In conclusion, the sometimes apparently conflicting data indicate the potential value of MCP-1, which would need to be assessed in more detail prior to clinical use.

#### 3.1.6. Matrix Metalloproteinase 7

Matrix Metalloproteinase 7 is a zinc-dependent endopeptidase upregulated in the kidney in acute or chronic kidney damage, transcriptionally activated by the WNT/β-Catenin pathway. Two independent studies showed the potential role of MMP-7 in AKI and CKD. In a cohort of 102 CKD patients, the urinary levels of MMP-7 were elevated in the CKD group compared to the healthy controls. MMP-7 was correlated with the degree of kidney fibrosis and inversely correlated with kidney function in patients with moderate CKD [40]. In a prospective multicenter cohort study on 721 patients (adults and children) undergoing cardiac surgery, urinary MMP-7 predicted moderate to severe AKI and was associated with a composite outcome for severe AKI, dialysis, and death, outperforming other biomarkers, including proteinuria or neutrophil gelatinase-associated lipocalin (NGAL), with an ROC-AUC of 0.81 in children and 0.76 in adults [41].

#### 3.1.7. Neutrophil Gelatinase-Associated Lipocalin

NGAL is a protein that was initially discovered in activated neutrophils, which then was shown to be produced in a variety of other cells, including kidney tubule cells, as a response to injury. NGAL has been shown to have predictive properties in AKI and, subsequently, evidence has been found for its importance in CKD, specifically in polycystic kidney disease and glomerulonephritis [42]. Four of the studies included in this review focused on the role of NGAL as a biomarker in DKD. The first study on 80 patients with type 2 diabetes, with a median eGFR of 92.4 mL/min/1.73 m² and median UACR of 4.69 mg/g, showed the urinary NGAL to creatinine ratio, among other markers, to be associated with albuminuria. NGAL was also correlated with diabetes duration. In a subgroup analysis and retrospective analysis, urinary NGAL, among others, was associated with the eGFR slope and changes in UACR [34]. In a cross-sectional study on 209 normoalbuminuric type 2 diabetes patients, the subgroup with eGFR < 60 mL/min per 1.73 m² had higher levels of urinary NGAL and NGAL was negatively correlated with eGFR. A multiple linear regression showed NGAL (β = −0 287, *p* =0008) to be independently correlated with eGFR [43]. In a retrospective study on 100 patients with type 2 diabetes and CKD, their urinary NGAL to creatinine ratios were assessed. Kidney biopsy results were available for the patients and allowed for them to be grouped into DKD and non-DKD. The urinary NGAL was significantly higher in the patients with DKD and urinary NGAL was an independent risk factor for DKD in the CKD patients with type 2 diabetes. Urinary NGAL showed, among others, correlations with proteinuria, eGFR, histological markers of inflammation, and CKD. In an adjusted model, urinary NGAL was associated with a higher probability of nephrotic-range proteinuria and lower event-free survival rates [44]. A prospective cohort study on type 2 diabetes patients with advanced nephropathy showed that patients in the highest quartile of urinary NGAL had a higher risk of reaching a composite outcome, including a rapid eGFR decline and ESKD, during a follow-up of 3 years [36]. In the above mentioned study by Brunner et al., in a prospective cohort of pediatric patients, urinary NGAL was also assessed and proved to be a moderate predictor of the histological features of kidney damage and a rapid eGFR decline, with a similar level of association as KIM-1, inferior to osteopontin and adiponectin [35]. 

#### 3.1.8. Uromodulin

Uromodulin (also known as Tamm–Horsfall protein) is produced in the kidney and physiologically excreted into the urine. It has been associated with immunological mechanisms and electrolyte balance and shown to have protective functions against urinary tract infections and kidney stone formation in animal knockout models. Several studies have investigated uromodulin as a serum marker for AKI and CKD [45,46,47,48,49,50]. It has been identified as a risk factor for CKD through GWAS and interest in its performance as a urinary biomarker has risen [51]. In a study on 364 patients who underwent kidney biopsies, urinary uromodulin levels were negatively associated with serum creatinine and patients with higher uromodulin levels had lower degrees of kidney fibrosis and glomerulosclerosis [52]. In the previously mentioned study by Puthumana et al., higher uromodulin levels were associated with a smaller eGFR decline and decreased risk of the composite kidney outcome. Combining urinary uromodulin with other biomarkers has improved its predictive performance [38]. In contrast to NGAL, urinary uromodulin could not be associated with eGFR, eGFR changes, or albuminuria, and its only significant association was with markers of diabetes control [34]. In a study on 101 patients who received cardiopulmonary bypass, preoperative levels of urinary uromodulin were inversely correlated with the incidence of AKI and urinary uromodulin strongly predicted postoperative AKI with an ROC-AUC of 0.90 [53].

#### 3.1.9. Other Single Biomarkers

84 patients who underwent kidney biopsies were assessed for serum and urinary growth differentiation factor–15 (GDF15), a member of the TGF-β superfamily, and followed-up for 29 ± 17 months. The urinary GDF15 levels were higher in patients with DKD. Urinary GDF15 was predictive of patient survival and a composite outcome of mortality and kidney replacement therapy with an ROC-AUC of 0.95 (95% CI 0.89–1.00, *p* < 0.001) [54]. A study on a TGF- β superfamily member, Activin A, was performed on 51 patients with ANCA-associated vasculitis (AAV), 41 of whom had kidney involvement. Interestingly, urinary Activin A was undetectable in healthy volunteers. Urinary Activin A was significantly increased in patients with kidney involvement compared to non-kidney AAV and correlated with other biomarkers of CKD such as proteinuria, liver-type fatty-acid-binding protein, and N-acetyl-beta-D-glucosaminidase. Furthermore, urinary Activin A was significantly higher in patients with a glomerular crescent formation (in a kidney biopsy), indicating ongoing glomerular inflammation and severe damage. After immunosuppressive treatment, urinary Activin A decreased rapidly [55]. A population of 70 biopsy-proven DKD patients with severely impaired kidney function and heavy proteinuria was tested for a panel of 10 selected urinary biomarkers. CX-C motif ligand 16 (CXCL16) and endostatin urinary levels were associated with the degree of kidney fibrosis and higher levels predicted a rapid loss of eGFR and poor kidney outcomes [56]. Serum Galectin-3 (Gal-3) levels have been associated with risks of incident CKD, rapid eGFR decline, and kidney fibrosis. Urinary Galectin-3 levels were investigated in a prospective cohort of 220 patients who underwent kidney biopsies. High urinary Gal-3 was associated with higher degrees of kidney fibrosis and a higher risk of CKD progression (adjusted HR, 4.60; 95% confidence interval, 2.85–7.71) [57]. Endotrophin, a factor released upon collagen VI deposition in the kidney, was tested as a urinary marker for CKD progression and kidney fibrosis in the Renal Impairment in Secondary Care (RIISC) cohort, a prospective observational study on 499 CKD patients. Urinary endotrophin levels were independently associated with a higher risk of CKD progression, improved a disease progression model, and were predictive of ESKD [58]. Other urinary markers of collagen VI and III formation and degradation (PRO-C6 and C3M) were assessed in a cohort of 663 patients with type 1 diabetes. Urinary levels of PRO-C6 were inversely correlated with an eGFR decline [59]. In two publications by connected groups of authors, a multibiomarker urinary assay, including cell-free DNA (cfDNA), methylated cfDNA, clusterin, CXCL10, total protein, and creatinine, was recently introduced. This so-called KIT Score was reported to have a good sensitivity and specificity for the detection of early-stage CKD (97.3% (95% CI: 94.6–99.3%) and 94.1% (95% CI: 82.3–100%)) in 397 of the 1169 recruited patients with various CKD stages [60]. In a second study, the KIT score performance was tested in a population of 34 IgA patients and 64 demographically matched healthy controls. An IgA-specific score was generated using the biomarker panel. The score was significantly higher for the IgA patients compared to that for the healthy controls (score value 87.76 vs. 14.03, *p* < 0.0001) and outperformed proteinuria [61].

### 3.2. Peptidomic/Proteomic-Based Biomarker Panels

As evident from the studies presented above, the accuracy and consequent reproducibility of findings based on single biomarkers are moderate, sometimes even with conflicting results. These issues, which also hold true for the classical biomarkers eGFR and albuminuria, have resulted in the concept of using multiple biomarkers to reduce variability and thereby increase precision. In the majority of peptide-based biomarker research studies on CKD, authors choose a panel of individual peptide biomarkers and combine them to a biomarker model. Such a combination of multiple peptide biomarkers increases stability and accuracy [62,63].

This emphasizes a potential benefit of choosing a large-scale, hypothesis-free approach when trying to discover the relevant biomarkers in certain populations or disease entities. Proteomic research offers high-resolution and high-throughput methods for identifying thousands of peptides within a specimen. The quantification and differential occurrence of peptide fragments can then be used to generate multidimensional classifiers containing up to hundreds of peptides differentially abundant between patient groups. Following this, we want to discuss the novel proteomic biomarkers and studies with impacts on CKD diagnosis, evaluation of progression, and the guidance of its treatment.

#### 3.2.1. CKD273

CKD273 is a urinary proteomic classifier containing 273 peptides that was originally discovered in 2010 [64]. It was derived from a human urinary database that contained, at that time, the urinary peptide data of 3600 patients analyzed using capillary electrophoresis coupled with mass spectrometry (CE-MS), a high-resolution, reproducible method for peptidome analyses. The diagnostic and prognostic properties of CKD273 in the various stages of CKD and numerous CKD etiologies, especially DKD, have been shown in a number of studies [65,66,67,68,69,70,71,72].

Recent studies on CKD273 have almost exclusively been focused on its predictive performance in patients with early-stage CKD, where the shortcomings of classical biomarkers such as eGFR and albuminuria limit their potential.

In a study by Pontillo et al., the question if CKD273 is superior to UACR in predicting CKD progression up to stage three (eGFR < 60 mL/min/1.73 m^2^) was raised. A total of 2087 individuals with an eGFR of >60 mL/min/1.73 m^2^ and minimal to normal albuminuria were included. Over a median follow-up of 4.6 years, CKD273 was superior to UACR in predicting a first and sustained renal endpoint [73]. This was also shown in a retrospective cohort of 1014 individuals with a baseline eGFR of ≥70 mL/min/1.73 m^2^ and urinary albumin excretions of <20 μg/min, showing the ability of CKD273 to identify the progression to eGFRs of <60 mL/min/1.73 m^2^ [74]. The risk stratification of eGFR loss in early-stage CKD was later improved by the generation of CKD273 sub-classifiers derived from the different eGFR strata within the entire patient cohort. Especially in patients without CKD or in early-stage CKD, these sub-classifiers outperformed albuminuria, the clinical Kidney Failure Risk Equation [75], and CKD273 [76]. In a smaller cohort of 155 type 2 diabetes patients with preserved kidney function and microalbuminuria, CKD273 showed correlations with eGFR and albuminuria. In a longitudinal follow-up, however, it failed to predict rapid eGFR loss and albuminuria. However, after multiple adjustments, CKD273 was a predictor for death but not for cardiovascular events in this cohort [77]. These findings raised the question of if screening patients with type 2 diabetes without known kidney impairments using CKD273 could be beneficial from an epidemiological and economic standpoint. Critselis et al. developed a decision analytic model evaluating individual costs and health outcomes, while hypothetically applying an annual CKD273 screening instead of albuminuria screening for these patients. The incremental costs exceeded the albuminuria screening, but the health benefits, including quality-adjusted life years, were predicted to outweigh the cost factors when focusing on high-risk patients [78]. Another study emphasizing the importance of CKD273 in the early CKD stages was published by Verbeke et al., including 451 patients with all the CKD stages over a median follow-up time of 5.5 years. They showed, after a multiple adjustment, that CKD273 was strongly predictive of fatal and non-fatal cardiovascular events in patients with early CKD stages and without an apparent history of cardiovascular events [79]. 

The performance of CKD273 in early-stage CKD led to a multicentre, prospective, observational study with an embedded randomized controlled trial (PRIORITY). The recruited patients had type 2 diabetes, a normal urinary albumin excretion, and preserved kidney function. The cohort was divided into a high- and a low-risk group according to their CKD273 scoring. The high-risk patients underwent placebo-controlled treatment, with 25 mg daily of spironolactone. The primary endpoint was the development of microalbuminuria. CKD273 proved to be predictive for the development of albuminuria; however, treatment with spironolactone did not significantly alter the progression course [80]. Similarly, CKD273 was able to predict microalbuminuria, independently of numerous other factors, including treatment with candesartan vs. a placebo in a large cohort of normoalbuminuric type 2 diabetes patients (Diabetic Retinopathy Candesartan Trials (DIRECT-Protect 2 study)) [81]. While no benefit of spironolactone treatment could be detected for early-stage DKD, the treatment response to spironolactone (reduction in UACR) had previously been demonstrated at more advanced-stage CKD. In this cohort of 101 patients with type 2 diabetes, the treatment response was predictable based on CKD273 [82]. 

#### 3.2.2. Other Biomarkers of DKD

In addition to CKD273, other proteomic-based biomarkers have been suggested for the assessment of DKD within the last 5 years. In a Taiwanese cohort of early-stage DKD patients, a proteomic approach was used to identify the candidate biomarkers, which were subsequently verified by an enzyme-linked immunosorbent assay. An analysis of a total of 114 patients led to the identification of candidate biomarkers, eight of which could be validated. This ultimately led to the identification of haptoglobin as a urine biomarker for early DKD detection and the prediction of early decline in kidney function [83]. In another study based on liquid chromatography–mass spectrometry (LC-MS), the authors sought to identify a peptide panel for a differentiation of the severity of DKD in 60 patients with different levels of albuminuria. The generated panel included collagen fragments and alpha-1 antitrypsin among the differentially occurring urinary peptides, similar to the observations with the classifier CKD273 [64,84]. Using LC-MS to identify the potential biomarkers of DKD in a retrospective study, 54 patients were grouped according to kidney outcomes and a urinary analysis was performed, which led to the identification of 66 peptides with differential abundances between the two groups. A combination of 5 of the 66 peptides was superior to albuminuria or eGFR in predicting kidney outcome [85].

#### 3.2.3. Biomarkers of Kidney Fibrosis

Kidney fibrosis is a hallmark of progressive disease in virtually all entities of CKD [86]. As of now, the degrees of interstitial fibrosis and tubular atrophy (IFTA) can only be assessed by invasive kidney biopsies, which come along with a number of problems and limitations, as outlined above. Several single peptide biomarkers have been proposed for a non-invasive estimation of kidney fibrosis [17,40,52,56,57]. In recent years, novel proteomic biomarkers reflecting the level of kidney fibrosis have been generated. CKD273 was correlated with the biopsy-proven degrees of kidney fibrosis in a cohort of 42 patients, whereas UACR and eGFR showed no association with fibrosis. This led to the identification of seven differentially abundant, fibrosis-associated peptides. All of the peptides were collagen fragments and displayed significant and negative correlations with the degree of kidney fibrosis, highlighting the role of collagen in the accumulation of the extracellular matrix, a hallmark of fibrosis [87]. In a subsequent study using CE-MS, with a larger cohort of 435 patients with various etiologies of CKD, a proteomic classifier containing 29 differentially excreted fibrosis-associated peptides (FFP_BH29) could be identified. The classifier was able to distinguish between patients with and without kidney fibrosis, with an ROC-AUC of 0.840 (95% CI: 0.779 to 0.889, *p* < 0.0001), and was significantly correlated with the degree of IFTA [88]. A large study focusing on collagen alpha 1(I) (col1a1) identified 501 different col1a1 fragments in the urine of 5000 patients with and without CKD. The vast majority of the differentially expressed fragments were positively correlated with eGFR and negatively correlated with ageing. The authors suggested that kidney fibrosis may be a consequence of decreased collagen degradation, rather than increased synthesis [89].

#### 3.2.4. Biomarkers in Different CKD Entities

IgA nephropathy (IgAN) is the most common primary glomerulonephritis and is characterized by a wide range of progression rates [90]. Therefore, risk stratification is highly relevant to identify individuals more likely to rapidly progress towards ESKD, in order to offer tailored therapies. Using an analysis of 209 patients’ urine samples via CE-MS, 237 peptides were identified that showed significantly different abundances in fast-progressing IgAN vs. slowly progressing IgAN. These peptides included, among others, fragments of apolipoprotein C-III, alpha-1 antitrypsin, different collagens, and uromodulin. A classifier based on these 237 peptides showed a significantly added value to clinical parameters for a prediction of IgAN progression [91]. In the context of SLE and LN, Pejchinovski et al. developed a panel of 65 urine peptides, including uromodulin and fibrinogen alpha, which was able to discriminate between SLE patients and healthy controls. The classifier was shown to identify patients with LN in a validation cohort with an ROC-AUC of 0.80 (*p* < 0.0001, 95%-CI 0.65–0.90) [92]. Autosomal dominant polycystic kidney disease (ADPKD) is a genetic disease characterized by bilateral kidney cyst formation and progression into ESKD. A proteomic biomarker panel containing 20 urinary peptides was able to predict rapid eGFR decline and an in silico analysis of cleavage sites revealed the potentially involved proteolytic pathways, including matrix-metalloproteinases and cathepsins, suggesting that altered proteolytic pathways are a part of disease progression [93]. In Fabry’s disease, a rare multisystemic disease with kidney involvement, a proteomic analysis was used to identify the different urinary biomarkers associated with asymptomatic, pre-symptomatic, and symptomatic disease, as well as kidney involvement [94]. In another small cohort for a different rare disease, Bardet-Biedl syndrome, proteomic profiling displayed 42 differentially occurring urinary peptides mostly involved in fibrosis and extracellular matrix organization [95]. A differential diagnosis of minimal change disease and focal segmental sclerosis was addressed in a proteomic approach with an ELISA validation, revealing a panel of biomarkers enabling discrimination between these hardly distinguishable CKD etiologies [96]. In a proteomic analysis of 120 patients with LN, uEGF was significantly associated with disease activity, histopathological findings, and CKD progression [97]. 

Diagnoses of CKD etiologies currently need a combination of clinical data, biomarkers, and foremost, kidney biopsies. In a recent study, the discrimination of seven different CKD etiologies from healthy controls and from each other was achieved through proteomic classifier development. In a cohort of 1180 patients, seven proteomic classifiers were developed to specifically differentiate MNGN, FSGS, LN, Vasculitis, IgAN, MCD, and DKD. The generated classifiers led to ROC-AUCs for discriminating these disease entities from others that ranged from 0.82 in IgAN to 0.95 in vasculitis-associated kidney diseases [98]. Furthermore, proteomic research was shown to enlighten the complement involvement in different CKD etiologies [99] and protease involvement in nephrotic syndrome [100] and other CKD entities [101]. It has furthermore improved our understanding of kidney peptide handling through comparisons of serum and urinary peptides in matched samples [102].

## 4. Discussion

In this review, we systematically searched the literature for frequently cited original articles published within the last 5 years that presented evidence for urinary peptide biomarkers in the context of CKD. In the first section, single peptide biomarkers are presented, followed by a section on proteomic biomarkers.

Recent studies on CD80 as a urinary biomarker have provided new evidence for its potential use as biomarker in the context of nephrotic syndrome. Its ability to differentiate MCD from other causes of nephrotic syndrome has been especially underlined. Another study from 2018, which did not reach the needed number of citations for this review, focused on its ability to differentiate MCD in relapse from FSGS and did not reach the conclusion that it has sufficient biomarker properties in this regard [103]. This is in contrast to prior findings, where CD80 was associated to MCD in relapse but not MCD in remission or FSGS [15]. These findings are in consensus with other studies showing the same association with good sensitivities, specificities, and AUCs [13,104,105,106]. Conflicting data might be due to the relatively low patient numbers in most of the trials, which is owing to the low incidences of the corresponding diseases, with the highest number of MCD patients included in a study being in the Mayo Clinic and NEPTUNE study by Gonzalez Guerrico et al. [13].

DKK3, as a marker of tubular damage, is described in this review and has been shown to have implications as a biomarker for CKD and kidney damage in the context of diabetes, after cardiac surgery, and in contrast-media-associated kidney injury [19,20,21,22,23,24]. Interest in DKK3 as a urinary biomarker has risen after a study by Federico et al., where the authors showed that the inhibition of DKK3 in mice led to reduced kidney fibrosis and that the urinary DKK3 levels were increased in patients with higher levels of interstitial fibrosis and tubular atrophy, possibly reflecting kidney damage [17]. These recent studies have led to DKK3 becoming a commercially available biomarker. Although some studies have investigated its properties as a plasma biomarker [107,108], its potential as a urinary biomarker seems to be higher. This may be due to it being secreted into the urine under tubular stress, thus directly reflecting ongoing tubular damage. There is hope that DKK3 might serve as a biomarker for other entities such as MCD or autosomal polycystic kidney disease as well, as pathophysiological connections have been made [109]. For now, its role in CKD and AKI remains more promising and requires further investigation.

As EGF is virtually absent in the plasma, there is no question about it being a urine-exclusive biomarker. The studies included in this review mainly provided evidence for its implications in reflecting kidney damage and predicting kidney function decline in DKD, after cardiac surgery and in Alport syndrome [26,27,28,29,30,31]. An upregulation of the renal EGF-receptor and a reduction in urinary EGF have been shown in a variety of animal models with kidney injury [110]. In a study by Betz et al., uEGF had already been shown to be predictive of kidney function decline, superior to albuminuria, in 642 diabetic normoalbuminuric individuals with preserved kidney function [111]. The predictive power of urinary EGF was improved by combining it with MCP-1 into a uEGF/uMCP-1 ratio, which had priorly been shown to have predictive values in IgA nephropathy and for kidney fibrosis in primary glomerulonephritis [112,113]. The precise role of EGFR activation and the role of EGF as a biomarker in different scenarios still has open questions, which might have to be answered before the clinical implementation of the biomarker can be targeted [114,115]. 

The transmembrane protein KIM-1 was shown to predict kidney function decline and reflect albuminuria in different cohorts of DKD in this review [26,34,36]. Furthermore, it appears to be predictive of histological damage in patients with LN [35]. KIM-1 was first cloned in different species by Han et al. in 2002. The authors linked the presence of the protein in the urine to acute tubular necrosis and, more specifically, proximal tubule damage [116]. Due to its absence in the urine of healthy individuals and seemingly good biomarker properties, as well as genomic findings, which indicate an enormous upregulation of KIM-1 in damaged kidneys, KIM-1 has gained a lot of scientific interest in the past two decades [117,118]. Subsequently, it has been studied in a variety of different entities of acute and chronic kidney damage, among others, cadmium nephrotoxicity, renal cell carcinoma, or as a marker of cellular injury in renal graft failure [119,120,121]. Recent advances have tried to include KIM-1 in biomarker panels, which might be promising in the future.

The chemotactic cytokine MCP-1 was extensively reviewed here, with evidence for its properties as a biomarker in DKD, after cardiac surgery, and in LN [26,37,38,39]. A combination with urinary EGF seems to greatly improve its predictive properties, with EGF being a marker for tubular damage and MCP-1 reflecting ongoing inflammation, which is a key feature of certain CKD etiologies such as LN, where the early detection of ongoing inflammation through regular screening is crucial for preserving kidney function and avoiding ESKD. Extensive reviews of MCP-1 and its potential role as a biomarker for kidney injury have been undertaken in the past [122,123]. After its discovery, it was quickly shown to be upregulated during inflammation, especially in DKD, and it was then proposed a biomarker of ongoing inflammation and a potential therapeutic target in DKD [124]. However, MCP-1 seems not to be restricted to DKD, as other reports have shown it to be associated with, for example, active renal vasculitis or cardiovascular events in CKD [125,126]. These reviewed findings, combined with prior studies, indicate that urinary MCP-1 seems to be a decent marker for active, inflammatory kidney disease and might also help to monitor treatment responses [39].Other markers might be more suited to the correct diagnosis of specific CKD etiologies or progress predictions than MCP-1.

MMP-7 is a secreted, zinc-dependent endopeptidase that is implicated in the regulation of kidney homeostasis and disease. Reviews about the regulation, role, and mechanisms of MMP-7 in the pathogenesis of kidney diseases have recently been provided [127]. In this review, we cite one smaller study on CKD and a fairly large cohort of AKI patients investigating MMP-7 as a urinary biomarker [40,41]. The first evidence that MMP-7 might play a role in tubular damage and thus be a candidate biomarker for kidney damage was collected in 2004 [128]. Furthermore, a pathophysiological link to kidney fibrosis through Wnt signalling has been made [129]. Unfortunately, only singular observations, such as MMP-7 as a serum marker for kidney damage and a recent study proposing it as an early urinary biomarker for kidney decline in hypertensive patients, have been made [130,131]. In summary, evidence for MMP-7 as a biomarker for kidney damage is present and has increased in the last 5 years. However, to clearly identify its true potential, more studies are needed.

Perhaps one of the most promising biomarkers regarding this review, but also historically, is NGAL, which was identified as a urinary biomarker for ischemic kidney damage roughly 20 years ago [132]. A multitude of studies followed this, enlightening its role in many entities with acute and chronic kidney damage, such as cardiac surgery, coronary interventions, platin toxicity, renal transplants, and more [133,134,135,136,137,138], but mainly focusing on its role as a marker for acute damage. In this review, clear evidence for its role as a urinary biomarker is given, especially in the context of chronic damage in DKD [26,34,35,36,43,44]. The reviews regarding the role of NGAL in kidney damage and evaluating its role as a biomarker in AKI and CKD are rather old and need revision in light of recent findings [42,139]. 

Uromodulin has been described as a promising serum biomarker for AKI and CKD [45,46,47,48,49,50]. In this review, evidence for its properties as a urinary biomarker is given in single peptide studies and studies focusing on a biomarker panel, with varying results and evidence that is not convincing [28,34,52,53]. In addition, older evidence indicates that uromodulin might better serve as a serum biomarker for CKD [140,141], although urinary biomarker properties find singular evidence every once in a while and might be still worth looking at more closely [142,143].

Most of the reviewed single peptide biomarkers had been described in the past and recent studies have only delivered mostly confirmatory evidence of known biomarker properties and limitations. The new discovery of single peptide biomarkers in the context of CKD has slowed down and biomarker discovery has shifted to high-throughput methods, which are able to analyze a multitude of peptide fragments in specimens, namely proteomics. 

Evidence for proteomic-based biomarkers is reviewed in this publication. CKD273 and IgAN237 have been extensively evaluated in prospective studies over the last 5 years [72,73,76,77,79,80,91]. Furthermore, proteomic DKD classifiers [83], classifiers for the differentiation of CKD etiologies [98], and fibrosis classifiers [87,88] have been published more and more frequently in the last 5 years.

This article and the biomarkers discussed within have limitations. First, the time frame selected for biomarker revision is rather short and thus the degree of evidence for every biomarker must be read with caution, as important preceding studies that did not fall in the search time frame might not be acknowledged. Secondly, we focused on urinary biomarkers specifically, and some of the mentioned biomarkers might have a higher importance and benefit as serum biomarkers. The heterogeneity of the biomarker studies selected in this article was high. Even if only human, clinical studies were included, and the patient numbers, biomarker assays, and definition of outcomes varied immensely across the studies. That was the main reason that we specifically decided not to perform a meta-analysis. Even though urine is a stable, easy-access, and reliable biomarker specimen, longer storage times, times of collection, bacterial contamination, and the need for time correction (creatinine correction) are factors that can limit the quality of urine specimens and thus influence the biomarker quality.

## 5. Summary and Conclusions

Urine is an easily and readily available specimen that allows unlimited and longitudinal sampling and testing. The advantage of urinary proteins and peptides is their stability [144]. Moreover, most urinary peptides are thought to originate from the urinary tract (especially from the kidney and bladder), thus making urine the ideal biomarker fluid for studying CKD. Within the last five years, substantial efforts have been reported on urinary peptidomic and proteomic biomarker research for CKD, with a resulting wealth of publications. Based on the reviewed articles, a list of the most relevant biomarkers or biomarker panels was generated. It summarizes the context of use for every identified biomarker, as well as the biomarker specificity for CKD stages and CKD etiologies, while analyzing the degree of evidence for every biomarker found within the last 5 years (Table 1).

Single-peptide-based biomarker research has been the state of art in biomedical research for decades. It requires a predefined pathophysiological hypothesis and has led to the discovery of many important biomarkers in the past. In the last 5 years, known urinary biomarkers have been addressed in single-peptide-based trials, but the discovery of new biomarkers has fallen behind. Proteomic biomarker studies offer a new hypothesis-free tool for biomarker discovery in biomedical research. Through an analysis of a multitude of potentially relevant peptides and a combination into multidimensional classifiers, they better reflect the complex nature of diseases such as CKD. These studies can offer new insights into the disease mechanism, diagnosis, prediction of disease progression, and treatment response. As discussed, the first important steps, such as the differentiation of different forms of kidney disease, prediction of the degree of kidney fibrosis, and prediction of progressive kidney disease, have been undertaken successfully.

Urinary peptidomic and proteomic biomarkers hold the promise to significantly improve CKD management in the future. The next essential step is to move from discovery to application and demonstrate the value of urinary proteins and peptides in guiding interventions in the context of CKD, in order to make an impact on patient management. Since CKD is a risk factor for cardiovascular disease (CVD), the ultimate goal of biomarker development is to ensure a reduction in the progression of early towards established CKD and CVD and in preventing target organ damage. The analysis of the entirety of urinary peptides/proteins provides the possibility of distinguishing and studying multiple kidney pathologies via an analysis of only one urine sample [145]. A schematic depiction of a possible application of these urine peptide/protein biomarkers is shown in Figure 3. An early detection, as well as a more precise definition of the underlying etiology, will allow for appropriate and early intervention. Patients with high-risk profiles could be regularly tested using specific proteomic profiles tailored for the early detection of CKD, coronary artery disease (CAD), and heart failure (HF). Early-detection proteomic biomarkers could precede clinical diagnoses and thus ensure earlier interventions, as showcased in the scheme in Figure 3. Such an application of urinary peptide/protein biomarkers could significantly improve current patient management, reduce CKD progression, and increase patients’ life expectancy and quality of life.

## Figures and Tables

**Figure 1 ijms-24-09156-f001:**
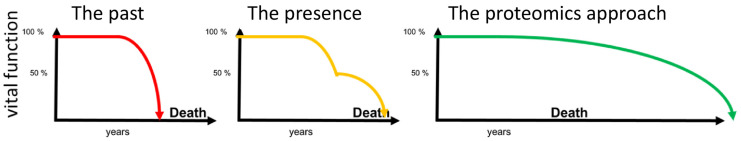
The promise of proteomics in the management of CKD patients. In the past (indicated in red), individuals died quickly from diseases, the transition from healthy to sick to death occurred fast. Current (indicated in yellow) patient care aims towards delaying death—but not preventing disease; treatment at late time point cannot stop or heal irreversible organ damage. Proteome analysis (indicated in green) enables early detection of cellular disease-specific changes, as the diseases are the results of proteome changes; the timely detection and early intervention allow prevention of the effects of diseases prior to irreversible organ damage.

**Figure 2 ijms-24-09156-f002:**
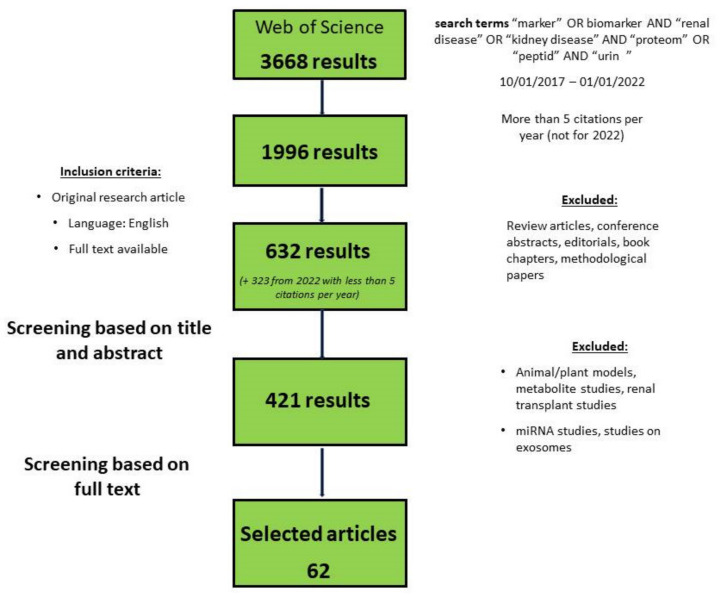
Flow diagram for the literature search strategy.

**Figure 3 ijms-24-09156-f003:**
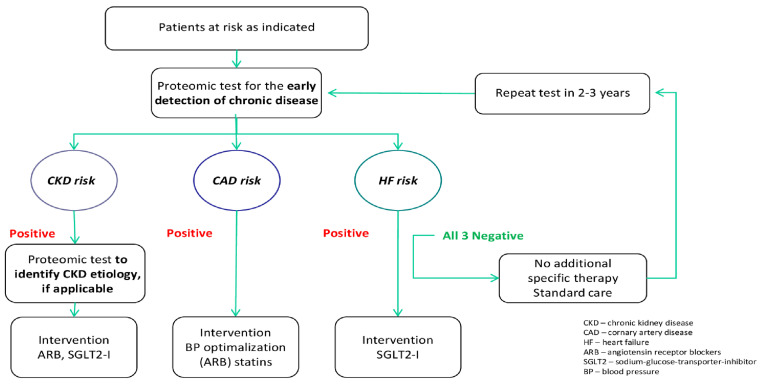
**Schematic depiction of peptide/protein biomarker application and the possible intervention consequence.** In case of positive results in the biomarker-scoring additional diagnostics workup, if applicable and treatment including management of risk factor according to the guidelines is suggested. In case of negative results, the monitoring within 2 to 3 years is proposed.

**Table 1 ijms-24-09156-t001:** Relevant urinary peptide and proteomic biomarkers of CKD.

Biomarker	Potential Context of Use	Studies/References	Evidence as Biomarker for CKD	Early/Late CKD Biomarker	Primary Diagnostic Biomarker Property	CKD Etiologies for Biomarker Use (If Specific)
CD80	Differentiation of MCD and FSGS; Response to immunosuppressive treatment in nephrotic syndrome; CKD progression in nephrotic syndrome	Gonzalez Guerrico et al. [13]; Ling et al. [14]; Garin et al. [15]; Ling et al. [16]	Intermediate	Late	Differential diagnosis; estimating CKD progression	Nephrotic syndrome; MCD; FSGS
DKK3	Association with CKD diagnosis, CKD progression, proteinuria; association with IFTA; Determinant of AKI in cardiac surgery; potential biomarker in contrast-media associated kidney injury	Sánchez-Álamo et al. [19]; Zewinger et al. [20]; Schunk et al. [21]; Rudnick et al. [22]; Seibert et al. [23]; Roscigno et al. [24]	High	Early	CKD detection; estimating CKD progression	CKD after cardiac surgery; contrast-media-associated kidney injury
EGF	Association with occurrence of DKD, eGFR and eGFR slope in DKD; potential biomarker of disease progression in Alport syndrome	Wu et al. [26]; Menez et al. [27]; Norvik et al. [28]; Satirapoj et al. [29]; Li et al. [30]	High	Early/Late	Differential diagnosis; estimating CKD progression	DKD; Alport syndrome
KIM-1	Biomarker in DKD, associated with UACR, eGFR and eGFR loss; biomarker for progression of albuminuria in early-stage DKD; possible biomarker for histological damage in LN	Gohda et al. [32]; Żyłka et al. [33]; Brunner et al. [34]; Satirapoj et al. [35]	Intermediate/partly Conflicting	Early	Differential diagnosis; estimating CKD progression	DKD; LN
MCP-1	Association with CKD incidence, progression, ESKD and death; potential biomarker in DKD; after cardiac surgery and development of CKD; in LN for kidney involvement, histological damage and response to treatment	Wu et al. [26]; Menez et al. [27]; Satirapoj et al. [29]; Puthumana et al. [37]; Davies et al. [38]	Conflicting	Late	CKD detection; estimating CKD progression; treatment response; differential diagnosis	DKD; CKD after cardiac surgery; LN
MMP-7	Biomarker of CKD, association with kidney fibrosis; Biomarker for AKI after cardiac surgery	Zhou et al. [39]; Yang et al. [40]	Low	Early/Late	Fibrosis marker; CKD detection	CKD after cardiac surgery
NGAL	Multiple evidence as biomarker for DKD; biomarker for AKI after cardiopulmonary bypass	Żyłka et al. [33]; Brunner et al. [34]; Satirapoj et al. [35]; Li et al. [42]; Duan et al. [43];	Conflicting	Early	Differential diagnosis; estimating CKD progression	DKD; CKD after cardiac surgery
Uromodulin	Inversely correlated with serum creatinine and kidney fibrosis and glomerulosclerosis in CKD; Potential protective effect and correlation with better outcome	Menez et al. [27]; Żyłka et al. [33]; Melchinger et al. [51]; Bennet et al. [52]	Intermediate	Late	CKD detection; estimating CKD progression	
GDF15	Association with DKD; Predictive of kidney replacement and survival	Perez-Gomez et al. [53]	Low	Late	Estimating CKD progression	DKD
Activin A	Association with kidney involvement in AAV; correlation with histological damage; decrease as potential marker of treatment response	Takei et al. [54]	Low	Late	Estimating CKD progression; treatment response	AAV
CXCL16	Predictive of fibrosis, rapid eGFR loss and poor kidney prognosis in cohort of advance CKD	Lee et al. [55]	Low	Late	Fibrosis marker; estimating CKD progression	
Galectin-3	Association with kidney fibrosis and CKD progression	Ou et al. [56]	Low	Late	Estimating CKD progression, fibrosis marker	
Marker of collagen formation and degradation	Collagen VI deposition marker associated with CKD progression and ESKD; Collagen formation marker inversely correlated with eGFR decline	Rasmussen et al. [57]; Pileman-Lyberg et al. [58]	Intermediate	Late	Estimating CKD progression; fibrosis marker	
CKD273	Peptide-based biomarker panel superior to UACR in predicting progression in DKD; sub-classifier for better risk stratification in early stages of DKD or healthy individuals; predictor for death and cardiovascular events in subgroups; predictive value confirmed in RCTs PRIORITY and DIRECT-Protect-2	Pontillo et al. [72]; Zürbig et al. [73]; Rodríguez-Ortiz et al. [75]; Currie et al. [76]; Verbeke et al. [78]; Tofte et al. [79]; Lindhardt et al. [80]	High	Early/Late	CKD detection; estimating CKD progression	DKD
FFP_BH29	Peptide-based biomarker panel for estimation of degree of kidney fibrosis	Catanese et al. [87]	Intermediate	Late	Fibrosis marker	
IgAN237	Peptide-based biomarker panel with significant added value regarding prediction of progress in IgA nephropathy	Rudnicki et al. [90]	Intermediate	Early/Late	Estimating CKD progression	IgAN
65SLE	Peptide-based biomarker panel for early diagnosis of SLE patients	Pejchinovski et al. [91]	Intermediate	Early	Differential diagnosis; CKD detection	SLE
ADPKD biomarker model	Peptide-based biomarker panel predicting relevant clinical outcomes in ADPKD patients a	Pejchinovski et al. [92]	Intermediate	Late	Estimating CKD progression	ADPKD
MCD vs FSGS urinary proteins biomarkers	Biomarkers differential diagnosis of MCD and FSGS	Pérez et al. [95]	Intermediate	Late	Differential diagnosis	MCD; FSGS
CKD differential diagnosis peptide panels	Biomarkers differential diagnosis of DN/Nephrosclerosis, IgAN, MN, LN, Vasculitis, MCD, and FSGS	Siwy et al. [95]	Intermediate	Late	Differential diagnosis	

High: significant association in ≥3 independent studies; moderate: significant association in 2–3 independent studies; low: significant association in one study; and conflicting: unresolved disparities in independent reports—no utility as a biomarker is implied. Abbreviations: AAV, ANCA-associated vasculitis; ADPKD, autosomal dominant polycystic kidney disease; AKI, acute kidney injury; CKD, chronic kidney disease; DKD, diabetic kidney disease; DKK3, Dickkopf-related protein 3; EGF, epidermal growth factor; eGFR, estimated glomerular function; ESKD, end-stage kidney disease; FSGS, focal segmental glomerulosclerosis; IgAN, IgA nephropathy; IFTA, interstitial fibrosis and tubular atrophy; KIM-1, kidney injury molecule-1; LN, lupus nephritis; MCD, minimal change disease; MN, membranous nephropathy; NGAL, neutrophil gelatinase-associated lipocalin; and SLE, systemic lupus erythematodes.

## Data Availability

Not applicable.

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
