# Peer review of "Recent Advances in Urinary Peptide and Proteomic Biomarkers in Chronic Kidney Disease: A Systematic Review"

_ijms, 2023, doi:10.3390/ijms24119156_

Round 1

Reviewer 1 Report

The work by Catanese and colleagues is a well written review which cleary describes the limitations of current parameters used in diagnosis and management of chronic kidney disease  and summarizes the recent findings in urinary peptide and proteomic biomarkers.

I have only few suggestions:

1.     I suggest to describe how the authors performe the analysis reported in fig.1.

2.     Please describe better the figure3 in the text.

Author Response

The work by Catanese and colleagues is a well written review which cleary describes the limitations of current parameters used in diagnosis and management of chronic kidney disease  and summarizes the recent findings in urinary peptide and proteomic biomarkers.

I have only few suggestions:

  1. I suggest to describe how the authors performe the analysis reported in fig.1.

We are uncertain as to what the reviewer wants to suggest. Figure 1 is a graphic depiction of the past, current, and future (expected) status of CKD management. No analysis was performed.

  1. Please describe better the figure3 in the text.

We expanded the description for this figure in the text and hope this meets the expectations of the reviewer.

Reviewer 2 Report

Catanese et al. collected all biomarker-related articles on chronic kidney disease (CKD) over the past five years, screened them, and finally selected 61 articles and 19 biomarkers for in-depth discussion. Therefore, this article summarizes all commonly used biomarkers for CKD in the past five years, providing readers with useful biomarker information. Importantly, all biomarkers discussed in this article have clinical data support, making it a valuable review article that provides valuable information for both clinical research and mechanistic experiments. The materials and methods used in this article are reasonable since it mainly consists of clinical articles and is recent. Moreover, as the article did not involve many statistics, the data presented are reliable. Therefore, the limitations of this article are minimal:

1. Although the topic of this article is to summarize biomarkers for CKD, many of these biomarkers are for diabetic kidney disease (DKD). Although the author separately introduced biomarkers for DKD and fibrosis in the result section, it is recommended that the author classify the 19 key biomarkers in more detail.

2. The author should discuss whether these biomarkers can be used to differentiate different types of diseases such as CKD, DKD, DM, HTN, etc. If possible, these biomarkers will be more valuable.

3. Some biomarkers, such as KIM-1 and NGAL, have significant changes only in acute kidney injury (AKI) or early-stage CKD, so it is inaccurate to describe them as CKD biomarkers in general. Therefore, it is recommended that the author differentiate all biomarkers into early-stage and late-stage CKD.

4. It is suggested that the author discuss the limitations of biomarkers and the limitations of this article in the discussion section.

Author Response

Reviewer 2:

Catanese et al. collected all biomarker-related articles on chronic kidney disease (CKD) over the past five years, screened them, and finally selected 61 articles and 19 biomarkers for in-depth discussion. Therefore, this article summarizes all commonly used biomarkers for CKD in the past five years, providing readers with useful biomarker information. Importantly, all biomarkers discussed in this article have clinical data support, making it a valuable review article that provides valuable information for both clinical research and mechanistic experiments. The materials and methods used in this article are reasonable since it mainly consists of clinical articles and is recent. Moreover, as the article did not involve many statistics, the data presented are reliable. Therefore, the limitations of this article are minimal:

  1. Although the topic of this article is to summarize biomarkers for CKD, many of these biomarkers are for diabetic kidney disease (DKD). Although the author separately introduced biomarkers for DKD and fibrosis in the result section, it is recommended that the author classify the 19 key biomarkers in more detail.

We thank the reviewer for this comment. We have tried to expand table 1 with also emphasizing the aspect mentioned in comment 3. We hope that a more detailed classification can be given this way. If further classification is necessary, we would like to ask the reviewer for suggestions of examples of this classification.

  1. The author should discuss whether these biomarkers can be used to differentiate different types of diseases such as CKD, DKD, DM, HTN, etc. If possible, these biomarkers will be more valuable.

Most of the mentioned biomarkers can either identify CKD, estimate progressions or help diagnose a specific CKD entity. We tried to show this in the results and also in the “potential context of use” column of table 1. We added a column to specifiy if differential diagnosis is possible and for what diagnosis for every biomarker. The biomarkers that can be used to differentiate different CKD entities are labeled “Differential diagnosis”.

  1. Some biomarkers, such as KIM-1 and NGAL, have significant changes only in acute kidney injury (AKI) or early-stage CKD, so it is inaccurate to describe them as CKD biomarkers in general. Therefore, it is recommended that the author differentiate all biomarkers into early-stage and late-stage CKD.

We found this comment to be very helpful and true. We expanded table 1 classifying every biomarker as early or late-stage CKD biomarker.

  1. It is suggested that the author discuss the limitations of biomarkers and the limitations of this article in the discussion section.

We thank the reviewer for this comment. We added a section discussing limitations of biomarkers and this article within the discussion section.

Round 2

Reviewer 2 Report

The authors have answered all my questions very perfectly. For instance, the "CKD etiologies for biomarker use (if specific)", "Early/Late CKD Biomarker", and "Primary diagnostic biomarker property" in table 1 are just the details I want the authors to provide.